# Extended Adjuvant Endocrine Therapy in Early Breast Cancer Patients—Review and Perspectives

**DOI:** 10.3390/cancers15164190

**Published:** 2023-08-21

**Authors:** Inga Bekes, Jens Huober

**Affiliations:** Breast Center, Kantonsspital St. Gallen, 9007 St. Gallen, Switzerland

**Keywords:** breast cancer, endocrine therapy, extended endocrine therapy, tamoxifen, aromatase inhibitor

## Abstract

**Simple Summary:**

Five years of therapy remains the standard for adjuvant endocrine therapy in early breast cancer. However, the recurrence risk remains elevated beyond this time period. Estimating the risk of recurrence as well as the efficacy of therapy is important in the selection of patients who will benefit from the extension of adjuvant endocrine therapy. The aim of this review is to summarize the major studies investigating the optimal duration of adjuvant endocrine therapy as well as to elaborate the possible individual indications for the extension of this therapy.

**Abstract:**

Seventy percent of all breast cancer subtypes are hormone receptor-positive. Adjuvant endocrine therapy in these patients plays a key role. Despite the traditional duration of a 5-year intake, the risk of relapse remains elevated in a substantial proportion of patients. Several trials report that the risk of late recurrence is reduced by the extension of adjuvant endocrine therapy beyond 5 years. However, the optimal duration of endocrine therapy is still a matter of debate. The newer data only show a marginal benefit resulting from extension beyond 7 to 10 years. Furthermore, extension may be associated with more side effects. Thus, the adequate selection of patients qualifying for an extended adjuvant therapy is of importance. Tools/genomic tests, which include the characteristics of the patient and the tumor, may help to better identify patients with a risk of a late relapse. Taken together, the magnitude of benefit for extended adjuvant endocrine therapy is based on the precise estimation of the risk of relapse after 5 years. This must be balanced against the long-term side effects of endocrine treatment and the competing risks. For patients with an intermediate risk, 7 years appears to be the optimal duration, and in those with high-risk features, endocrine therapy up to 10 years may be considered.

## 1. Introduction

The number of breast cancer survivors has been rising over the last 20 years. Currently, they form the largest group of cancer survivors [1]. With about 70% of all breast cancers, hormone receptor-positive, HER2-negative is by far the most common subtype [2]. Adjuvant endocrine therapy has been the cornerstone of systemic treatment for these breast cancer subtypes for many years. Taking tamoxifen for 5 years has been shown to significantly reduce the 15-year risk of recurrence by 40%, as well as the 15-year risk of mortality by about 30%, compared to patients receiving a placebo [3]. In the comparison of an endocrine therapy with an aromatase inhibitor to tamoxifen in postmenopausal women, the aromatase inhibitor was associated with about one-third fewer recurrences during the treatment period and with approximately 15% fewer deaths from breast cancer during the first decade [4]. Despite these effective 5-year therapies, the risk of tumor relapse remains increased for a relevant number of patients beyond the 5 years, with evidence of recurrences up to 20 years after the primary diagnosis. For this reason, there is a good rationale for extending adjuvant endocrine therapy beyond the 5 years of initial treatment. However, till now the optimal duration of endocrine therapy extension remains unclear. At the same time the duration of therapy is an important factor because adjuvant endocrine therapy may be associated with side effects like hot flashes, sexual dysfunction, weight gain, bone pain, bone density loss, arthropathy, depression, cognitive dysfunction, and fatigue [5,6,7,8,9]. Consequently, treatment adherence is compromised, and a substantial percentage of breast cancer patients finish treatment prematurely, resulting in worse outcomes [10]. The challenge is to balance potential benefits against possible additional side effects and, in this way, to identify the patients who would benefit from an extended adjuvant endocrine therapy.

Therefore, the focus of this review is to summarize the major studies investigating the optimal duration of adjuvant endocrine therapy and to elaborate the possible indications for the extension of this therapy.

## 2. Relevant Section

### 2.1. Risk of Relapse after 5 Years of Endocrine Therapy

In 2017, the Early Breast Cancer Trials Collaborative Group (EBCTCG) showed in a review of more than 91 studies, which included a total of 46,000 patients, that more than 50% of the breast cancer recurrences in hormone receptor-positive breast cancer patients occur after completion of a 5-year period of endocrine therapy [11]. Recurrences could be observed up to 20 years after the primary diagnosis. The risk of recurrence at 20 years varied between 14% and 47%. Positive lymph nodes, advanced tumor size and poor differentiation of the tumor were recognized factors associated with an increased risk of relapse. However, two-thirds of the included patients had received only tamoxifen as adjuvant endocrine therapy. Moreover, the HER2 receptor status was also unknown in 50% of the patients. Forty percent of the patients were diagnosed before the year 2000, with only thirteen percent of the patients being diagnosed after 2005, meaning that many patients in this meta-analysis were treated suboptimally according to today’s therapy criteria. Overall, it has to be assumed that the risk of recurrence was overestimated in this meta-analysis. Nevertheless, the results of this analysis confirm the need to discuss and adjust the duration and the choice of substance for adjuvant endocrine systemic therapy according to the patient’s individual risk profile.

### 2.2. Five-Year Duration of Endocrine Therapy—Current Data

Taking tamoxifen for 5 years has been shown to reduce the risk of breast cancer recurrence by 40% and mortality by 30% compared to patients taking a placebo [3]. The early results of a Swedish study published in 1998 were able to show that 5 years of tamoxifen was more effective than 2 years, with a statistically significant increase in event-free survival and overall survival [12]. For this reason, a 5-year therapy period with tamoxifen was the standard of care for a long time. Postmenopausal patients are generally offered an aromatase inhibitor as a monotherapy of adjuvant endocrine therapy. In the case of intolerance or contraindications to aromatase inhibitors, however, a 5-year tamoxifen therapy period could be proposed. In this clinical setting, aromatase inhibitors can be taken either continuously for 5 years (as “upfront” strategy) or for 2 to 3 years after 2 to 3 years of adjuvant endocrine systemic treatment with tamoxifen (as a “switch” strategy). In fact, in several analyses there were no relevant differences reported concerning survival outcomes with these two different regimes [4,13]. In the EBCTCG meta-analysis, patients with a luminal-like early breast cancer and different schedules of adjuvant endocrine therapy were compared. A 5-year “upfront” endocrine therapy with an aromatase inhibitor was associated with a lower risk of breast cancer recurrence (10-year recurrence risk 19.1% versus 22.7%), as well as lower cases of death (RR 0.85; 95% CI, 0.75–0.96, *p*-value = 0.010), compared to 5 years of tamoxifen. Taking aromatase inhibitors after tamoxifen (a “switch” strategy) reduced the recurrence rate significantly in comparison to 5 full years of tamoxifen (17% versus 19%); both the breast cancer-related mortality and the overall mortality were shown to be reduced (RR 0.84; 95% CI, 0.71–0.96; 2*p*-value = 0.015; (RR 0.82; 95% CI, 0.73–0.91; 2*p*-value = 0.0002) in favor of the “switch” strategy. Comparing the different therapy strategies (“upfront” versus “switch”), a modest, and yet significantly lower, recurrence risk in favor of the “upfront” strategy was demonstrated only in the first 2 years of adjuvant endocrine therapy (RR 0.74; 95% CI, 0.62–0.89; 2*p*-value = 0.002). Comparable recurrence rates were reported in the subsequent 3 years of endocrine therapy, when all the patients were receiving an aromatase inhibitor (RR 0.99; 95% CI, 0.85–1.15). A comparison of breast cancer-related mortality as well as overall mortality in both groups showed similar results [4]. The data of the FATA-GIM3 study, in which 3697 postmenopausal patients were randomized, confirmed the results obtained by the EBCTCG-meta-analysis. In this study, women with early breast cancer received either one of the aromatase inhibitors, anastrozole, exemestane, or letrozole, “upfront” for 5 years or a “switch” therapy regime of tamoxifen for 2 years and subsequently one of the three aromatase inhibitors for the following 3 years. In this analysis, disease-free survival did not differ between the two treatment groups at 5 years of median follow-up (88.5% versus 89.8% in the “switch” and “upfront” regimes (HR 0.89, 95% CI, 0.73–1.08; *p*-value = 0.23)). Moreover, no differences in overall survival were reported (95.3% versus 96.8% in the “switch” and “upfront” groups (HR 0.72. 95% CI, 0.51–1.00; *p*-value = 0.052)). Analysis comparing the three aromatase inhibitor regimes showed a comparable efficacy concerning disease-free survival and overall survival (DFS rates: 90.0% versus 88.0% versus 89.4%; OS rates 95.9% versus 95.7% versus 96.6% for anastrozole, exemestane and letrozole, respectively) [13].

### 2.3. Optimal Duration of Extended Adjuvant Endocrine Therapy: 5, 7 to 8, or 10 Years?

With the knowledge of a persistently increased risk of relapse beyond a therapy period of 5 years, an increasing number of studies were published evaluating the effect of extended endocrine therapy with regard to disease-free survival and overall survival (Table 1).

### 2.4. Ten Years versus Five Years of Endocrine Therapy

The ATLAS [14] and aTTom [15] studies from 2012 and 2013 examined the effect of extending tamoxifen intake for a further 5 years after taking tamoxifen for 5 years (with the absence of recurrence) versus a placebo. In these two large studies with more than 6000 study participants included in each trial, both post- and premenopausal patients could be included.

The ATLAS study randomized 12894 pre- and postmenopausal women with early breast cancer after 5 years of tamoxifen intake to either finish the endocrine therapy or to continue with tamoxifen for another 5 years. In 6846 women, the extended therapy was associated with a reduction in the breast cancer recurrence risk (RR 0.84; 95% CI, 0.76–0.94; *p*-value = 0.002), breast cancer mortality (331 versus 397 deaths, *p*-value = 0.01), and overall mortality (639 versus 722 deaths, *p*-value = 0.01). At the same time, an absolute increase in the endometrial cancer-related mortality risk of 0.2% was registered [14]. The study population included 630 premenopausal patients. In the subgroup analysis, the results did not differ. For those patients with continued ovarian activity, however, there was a small risk of uterine cancer as well as vascular side effects [14].

The aTTom trial included 6953 patients that had received endocrine therapy with tamoxifen for at least 4 years. The patients were divided into those that prolonged the therapy with tamoxifen for another 5 years and those that discontinued the therapy. The data showed that, at a median follow-up of 9 years, the extended therapy with tamoxifen was associated with a decrease in the recurrence rate (28% vs. 32%; RR for recurrence 0.85, 95% CI, 0.76–0.95; *p*-value = 0.003) as well as a non-significantly reduced breast cancer mortality rate. Concerning side effects, an increased risk for endometrial cancer (102 events versus 45 events; rate ratio 2.20, *p*-value= 0.02) was demonstrated for patients receiving prolongation of the therapy [15]. In both trials, the reduced risk of recurrence reported in women with extended adjuvant endocrine therapy was particularly evident starting from year 10 [14,15].

The MA.17 study [16] used a comparable study design, evaluating the use of the aromatase inhibitor letrozole versus a placebo in 5187 postmenopausal patients after initial treatment for 4.5 to 6 years with tamoxifen. The data showed a statistically significant improvement in disease-free survival for women with letrozole extension. This result was associated with substantial reductions in local, distant, and contralateral events. Overall survival did not differ in both arms (HR for death from any cause = 0.82, 95% CI = 0.57 to 1.19; *p* = 0.3). However, patients with positive nodal status showed a statistically significant improvement in the overall survival with letrozole (HR = 0.61, 95% CI = 0.38 to 0.98; *p* = 0.04). Concerning toxicity, the women receiving letrozole suffered from more hormonally related side effects than those in the placebo group, but the incidence of bone fractures and cardiovascular events was comparable [16].

The following studies evaluated the extension of adjuvant endocrine therapy (aromatase inhibitor versus placebo for an additional 5 years) after 5 years of endocrine therapy if an aromatase inhibitor was taken continuously or in sequence with tamoxifen within the first 5 years.

In the NSABP B-42 study, 3966 patients receiving either “upfront” therapy with an aromatase inhibitor or the “sequential” intake of tamoxifen and aromatase inhibitor were randomly assigned to another 5 years of letrozole or placebo. Disease-free survival did not differ significantly at a medium follow-up of 6.9 years, in favor of therapy prolongation (0.85 [0.73–0.99]; *p* = 0.048). The primary differences in the frequency of disease-free survival events between the placebo and letrozole groups were recognized in distant recurrence (28% reduction in the letrozole group compared to placebo) and contralateral breast cancer. However, a difference in distant recurrence events was not observed before 4.1 years. Concerning side effects, there was a moderately elevated 7-year cumulative incidence of osteoporotic fractures (placebo 4.8% versus 5.4% letrozole group). Moreover, arterial thrombotic events were significantly increased after 2.5 years of letrozole: the 2.5-year cumulative incidence of 0.9% in the letrozole group increased to 4.0% after 7 years [17].

The MA.17R trial randomized 1918 postmenopausal patients to another 5 years of aromatase inhibitor therapy or to a placebo after 5 years of “upfront” treatment with an aromatase inhibitor. Disease recurrence rates were significantly lower (by 34%) among the patients with extended endocrine therapy with an aromatase inhibitor for up to 10 years than they were among the patients who received a placebo. At a median follow-up of 6.3 years, overall survival showed no benefit in favor of the advanced therapy with an aromatase inhibitor. The significant benefit in disease-free survival includes not only a numerically larger reduction in events of local, regional, and distant recurrence but also an apparently greater proportional reduction in events of contralateral breast cancer, which may partly explain the absence thus far of an observed overall survival benefit. Concerning side effects, the overall incidence was comparable in both groups. However, bone-related toxicity was higher in the letrozole group. In both treatment arms, a similar number of patients used bone-protecting medications like bisphosphonates (46.2% in the letrozole groups versus 46.6% in the placebo group) [7].

### 2.5. Seven to Eight Years versus Five Years of Endocrine Therapy

Other studies investigated the effect of extended adjuvant endocrine therapy over the period of 7 to 8 years. The DATA and GIM4 trials randomized patients with aromatase inhibitor intake versus a placebo for 2 versus 6 years after initial tamoxifen therapy for 2 to 3 years [18,19]. The DATA trial evaluated the role of extended endocrine therapy in 1860 postmenopausal patients who were homogeneously given sequential tamoxifen and aromatase inhibitors. The study reported an HR of 0.79 (95% [0.62–1.02]; *p* = 0.066) for 5-year disease-free survival in patients given 6 years versus 3 years of anastrozole after 2–3 years of adjuvant tamoxifen. In an unplanned post hoc analysis alone, improved disease-free survival was seen for nodal-positive patients (absolute reduction 8.2%) as well as nodal-positive patients and patients with a tumor stage ≥ T2 (absolute reduction 13.5%) [18].

In contrast, the GIM4 trial, in which 2054 postmenopausal patients were enrolled, showed significant results in terms of disease-free survival (HR 0.78 [0.65–0.93], 12 y: 62% vs. 67%) and overall survival (HR 0.77 [0.60–0.98]; 12 y: 84% vs. 88%) after a long median follow-up of 140 months [19]. Concerning toxicity in both trials, the extended duration of an aromatase inhibitor was associated with an increased incidence of side effects such as bone-, joint-, and muscle-related complaints; however, in the GIM4 trial there was no difference in the incidence of bone fractures.

### 2.6. Ten Years of Endocrine Therapy versus Seven to Eight Years

The IDEAL [20] and the SALSA (ABCSG 16) [21] study examined the difference between prolonged endocrine therapy for 7 to 8 years versus 10 years in postmenopausal patients, respectively.

In the IDEAL trial, 1824 postmenopausal hormone receptor-positive breast cancer patients were randomly assigned, after 5 years of any endocrine adjuvant therapy, to groups receiving either 2.5 or 5 years of letrozole. Disease-free survival showed no significant difference between the groups after a median follow-up of 6.6 years (HR 0.92, 95% CI 0.74–1.16). Furthermore, no significant differences in overall survival or distant metastasis-free survival were recognized. However, there was a significant reduction in the occurrence of second primary breast cancer in patients with 5 years of extended treatment (HR 0.39, 95% CI 0.19–0.81). Overall, another 5 years of endocrine therapy with letrozole was not superior to 2.5 years of extended adjuvant endocrine therapy after an initial 5 years of therapy [20].

The SALSA (ABCSG16) trial included 3470 women with any type of adjuvant endocrine therapy for 5 years to receive further treatment with anastrozole for another 2 years or for another 5 years (a total of 7 or 10 years of endocrine therapy). At 10 years—with regard to randomization—there was neither a difference reported in this trial concerning disease-free survival (73.6% versus 73.9%, HR 0.99; 95% CI; 0.85–1.15; *p*-value = 0.90) nor one concerning overall survival (87.5% versus 87.3%, HR 1.02; 95% CI, 0.83–1.25) when comparing 7 versus 10 years of extended adjuvant endocrine therapy. However, in patients receiving 10 years of endocrine therapy more osteoporotic bone fractures were reported (HR 1.35; 95% CI, 1.00–1.84) [21].

### 2.7. Intermittent versus Continuous Extended Therapy

The SOLE trial enrolled 4884 women after adjuvant endocrine therapy for 4 to 6 years. The patients received letrozole either continuously for another 5 years or intermittently (every day for 9 months, followed by 3 months of suspension in the first 4 years and then every day for the entire fifth year) [22]. The study background was based on pre-clinical data that had demonstrated how the transient suspension of endocrine therapy allowed a delay in the onset of endocrine resistance and therefore a prolongation of the benefits of an endocrine therapy intake [23]. In this study, an intermittent administration of endocrine therapy did not improve the disease-free survival in comparison to a continuous intake (7-year disease-free survival rate was 81.4% versus 81.5%, HR 1.03; 95% CI, 0.91–1.17; *p*-value = 0.64). However, distant recurrence rates were not greater in patients with intermittent therapy in comparison to a continuous administration in the extended therapy regime group (7-year distant recurrence-free interval 91.6% versus 90.4%, HR 0.91. 95% CI, 0.76–1.10; *p*-value = 0.35) [24].

New study results have emerged from the prospective POSITIVE trial [25]. In this single-group trial, a temporary interruption of endocrine therapy was evaluated to attempt pregnancy in young patients who had previously had breast cancer. Five hundred and sixteen young women were enrolled to interrupt endocrine therapy for about 2 years after 18–30 months of treatment. The study results showed no greater short-term risk of breast cancer events, including distant recurrence when compared to a matched cohort from the SOFT and TEXT trial. However, the majority of patients included had stage 1 or 2 tumors. In this lower-risk population, endocrine therapy for only 5 years can be considered a reasonable option for many of them compared to a higher-risk population. Follow-up is currently short and, particularly in the hormone receptor-positive setting, a longer follow-up is essential since half of the relapses emerge late after 5 years. We think that, based on the data from the POSITIVE trial, the interruption of endocrine therapy mainly concerns patients with a low risk of disease and cannot actually be transferred to a higher-risk population.

### 2.8. Side Effects

While adjuvant endocrine therapy is mostly well tolerated, there are still side effects concerning the duration of use that need to be taken into consideration. Muscle and joint pain and cognitive impairment, as well as loss of libido and sexual health, can be a major burden on the individual patient’s daily life and can thus compromise the adherence to therapy. Bone health must also be included in the concept of endocrine breast cancer therapy, as endocrine therapy may lead to loss of bone density and structure. In 2020, Zhao et al. evaluated the side effects of extended adjuvant endocrine therapy compared to a 5-year duration of use in a large meta-analysis [26]. The combined study included 24,187 patients. Extended adjuvant endocrine therapy with aromatase inhibitors significantly increased the risks of side effects such as cardiotoxicity, bone pain, osteoporosis, fractures, hot flashes, arthralgia, and myalgia; the risk of hypertension, hypercholesterolemia, vaginal dryness, nausea, constipation, headache, dizziness, and dyspnea were—at the same time—not increased. Significance was only reported with regard to the risk of hot flashes at grade ≥3 but not with any of the other side effects analyzed in this study population [26].

However, symptoms may even occur without endocrine therapy. Placebo-controlled trials evaluating patient-reported outcomes [7,8,16,27,28,29,30] show that women in the placebo group reported more than half as many episodes as those in the endocrine therapy group. This highlights the importance of a placebo control to assess as many pharmacologic side effects as possible. Also, little difference in discontinuation rates between endocrine therapy and the placebo were recognized, although in clinical practice a substantial number of women discontinue adjuvant endocrine therapy prematurely because of toxicity [10,31]. It has been shown that women who have already finished 5 years of adjuvant endocrine therapy suffer less from unexpected symptoms than those who are initiating therapy. These findings underline the need to sensitize patients who are receiving endocrine therapy in order to distinguish between pre-existing symptoms and new ones (which are normally attributed to endocrine therapy, such as menopausal symptoms [32,33], musculoskeletal symptoms [34], sexual dysfunction [35,36], and osteoporosis [37]. Such patients could try stopping treatment for short periods or switching from one agent to another [10]. Switching between an aromatase inhibitor and tamoxifen, as well as between various aromatase inhibitors, has been shown to be safer than discontinuing therapy prematurely. An overview of side effects that were elevated in the different performed studies are summarized in Table 2.

### 2.9. Optimal Duration of Extended Endocrine Therapy

In patients with tamoxifen as the only endocrine therapy during the first 5 years, the extension to 10 years clearly favors 5 further years of endocrine therapy with both an aromatase inhibitor and 5 further years of tamoxifen. However, 5 years of tamoxifen alone during the first 5 years is today only a treatment option in a minority of patients with an absolutely low risk of disease who obviously do not need therapy beyond 5 years.

For all other patients, the optimal duration of extended endocrine therapy remains unclear. In patients who had an aromatase inhibitor-based endocrine treatment during their first 5 years of therapy, the extension of therapy to 10 years by 5 further years of an aromatase inhibitor (NSABP B-42, MA.17R) did not result in an increased overall survival, and differences in disease-free survival events in favor of longer treatment were often caused by contralateral breast cancers.

In contrast, trials investigating the extension of endocrine therapy from 5 to 7 to 8 years clearly favored longer treatment with regard to disease-free survival and overall survival. However, the trials further extending therapy from 7 to 8 to 10 years (IDEAL and ABCSG 16) failed to show a better outcome with the extension of endocrine therapy to 10 years. The risk profile may be one of the reasons, such as in the ABCSG 16 trial where 65% of patients with early breast cancer were node-negative and where 73% had T1 disease.

Nevertheless, it appears that for the majority of patients 7 to 8 years of endocrine therapy might represent the best balance between efficacy and side effects since in most of the trials longer treatment was not unexpectedly associated with more side effects, like bone and cardiovascular toxicity or menopausal symptoms.

Unfortunately, most of these trial data only apply to postmenopausal patients since, with the exception of the aTTom and ATLAS trials, all the other studies did not enroll premenopausal patients. Thus, for premenopausal patients there is only evidence from the aTTom and ATLAS trials, where the benefit of extending tamoxifen from 5 to 10 years was also seen in premenopausal patients. However, the best way to further treat premenopausal patients who had ovarian function suppression therapy combined with tamoxifen or an aromatase inhibitor during their first 5 years of endocrine therapy remains open and is an individual decision ranging from switching to tamoxifen or extending ovarian function suppression combinations with tamoxifen or an aromatase inhibitor.

### 2.10. Individualization of Extended Adjuvant Endocrine Therapy

Studies have shown that patients with a low absolute risk of disease recurrence generally also only have a low absolute benefit from extended adjuvant endocrine therapy. Therefore, an assessment of the individual benefit—including an estimation of the long-term recurrence risk as well as the treatment efficiency—is essential to identify patients with an expected benefit without causing unnecessary side effects and toxicities. The criteria for a benefit from extended adjuvant endocrine therapy may include an initial advanced tumor stage, such as positive nodal status, or a large primary tumor size. However, patients with more aggressive tumor biology, such as poor differentiation or high proliferation, also had an increased risk of recurrence even after 5 years in the EBCTCG meta-analysis [38].

To classify individual patients into a risk category (low-/medium-/high-risk), tools may be used for decision making in single cases. Genomic tests are extensively utilized in patients with hormone receptor-positive early breast cancer or limited nodal disease to determine whether they may benefit from additional adjuvant chemotherapy or whether they could be successfully treated with endocrine therapy only. The Clinical Treatment Score post-5 years (CTS5) [39] is a web-based calculator, which allows a simple and practical risk calculation for patients with regard to the occurrence of distant metastases after a recurrence-free interval of 5 years under adjuvant endocrine therapy. The calculation requires the patient’s age and tumor-specific data such as tumor size, grading, and the number of affected lymph nodes. Based on these characteristics, patients are divided into a low-risk group (5–10-year risk < 5%), an intermediate-risk group (5–10%), and a high-risk group (>10%). As a limitation, the calculation is only validated for postmenopausal patients with HER2-negative receptor status.

Another multigene signature, which can be used as a benefit predictor for extended adjuvant endocrine therapy in patients after completion of 5 years of adjuvant endocrine therapy is the Breast Cancer Index (BCI) [39]. The BCI uses formalin-fixed paraffin-embedded tumor tissue and includes a combination of two gene expression signatures—the Molecular Grade Index, which examines cell cycle-associated genes, and the HI ratio (HOXB13/IL17BR ratio), which assesses the response to endocrine therapy. The classification of patients into groups (BCI low/high) can be included in the decision making of an extended adjuvant endocrine therapy. In the TransATTOM study [40], as well as in the IDEAL [41] and the MA.17 study [42], a significant risk reduction by extended adjuvant endocrine therapy was demonstrated for the HI high-risk patients (absolute risk reduction between 10.2 and 16.5%), in contrast to the BCI low-risk patients, who did not benefit significantly from prolonged endocrine therapy. However, concerning those patients randomized in the IDEAL trial, only BCI (not CTS5) was predictive of a benefit from the extended strategy [39]. Updated ASCO guidelines mention these tests as a possibility for therapy decision making in patients with negative lymph node status or with less than four affected lymph nodes [39].

## 3. Role of Other Receptor-Targeting Agents—Current and Future Aspects

It has been shown that women with advanced hormone receptor-positive/HER2-negative breast cancer may develop resistance against aromatase inhibitors over time. This observation is frequently caused by an Estrogen Receptor 1 (ESR1)-mutation [43]. How these mutations affect tumor sensitivity to the established therapies, and novel therapies are currently active areas of research. These therapies include estrogen receptor-targeting agents, such as selective estrogen receptor modulators, covalent antagonists, and degraders (including tamoxifen, fulvestrant, etc.), and combination therapies (with CDK4/6, etc.). In January 2023, the FDA approved the first selective estrogen receptor degrader (SERD) elacestrant for postmenopausal women or adult men with hormone receptor-positive/HER2-negative, ESR1-mutated advanced or metastatic breast cancer with disease progression following at least one line of endocrine therapy. The efficacy was evaluated in the EMERALD trial (NCT03778931), a randomized, open-label, multicenter trial, in which 478 postmenopausal patients were enrolled [44]. EMERALD is the first phase III clinical trial to study an oral SERD. Elacestrant significantly reduced the risk of death or disease progression and lengthened progression-free survival compared with fulvestrant or an aromatase inhibitor. The results suggest that in the future SERDs such as elacestrant may become treatment options, not only in the advanced setting but maybe also in combination with other targeted therapies even in the early setting of breast cancer treatment.

Based on the data of the monarchE trial investigating the CDK4/6 inhibitor abemaciclib for 2 years in the early setting, abemaciclib was approved in combination with tamoxifen or as an aromatase inhibitor for the adjuvant treatment of women with node-positive early breast cancer at high risk of recurrence (≥4 positive lymph nodes or 1–3 positive lymph nodes and either tumor grade 3 or T3-tumors or high KI67) [45]. The combination therapy regime showed significant improvement in disease-free survival and distant relapse-free survival compared to endocrine therapy alone. The first preliminary results of the Natalee trial show similar results with the CDK4/6 inhibitor Ribociclib [46].

How far such targeted therapies like CDK4/6 inhibitors, which are already in clinical practice in a higher-risk population, or new SERDs, which may perhaps become part of endocrine therapy in the early setting, will influence the risk of later relapses and the efficacy of extended adjuvant endocrine therapy is presently unknown. It is necessary to await the longer follow-up of these trials with CDK4/6 inhibitors or new SERDs before drawing definitive conclusions.

## 4. Conclusions and Further Directions

For many patients, endocrine therapy for 5 years remains the standard of care, and it is sufficient. However, the risk of recurrence remains elevated beyond 5 years in early hormone receptor-positive breast cancer patients. Estimating the risk of recurrence as well as the efficacy of therapy is important in the selection of patients who will benefit from the potential extension of adjuvant endocrine therapy.

As most patients have in general already received aromatase inhibitor-based therapy in the first 5 years, the optimal duration of therapy is most likely 7 to 8 years for patients at an intermediate risk of recurrence (N0-N1). An endocrine therapy duration of 10 years may be considered for patients with a very high risk of relapse (e.g., primary extensive nodal involvement N2-N3) (Figure 1). Aromatase inhibitors should be part of the treatment strategy for the majority of patients (except very low-risk patients or those with contraindications to aromatase inhibitors). Depending on menopausal status, patients who received 5 years of tamoxifen may be considered for a 5-year extension of tamoxifen or an aromatase inhibitor. Finally, individual aspects such as treatment tolerability and comorbidities must be included in the decision in order to ensure treatment adherence and to maintain the patients’ quality of life. The clinical multivariate web tool CTS5, as well as the genomic multiparametric assay BCI, may contribute to individual decision making.

Competing risks like age and comorbidities and tolerability with regard to prior therapy need to be considered.

## Figures and Tables

**Figure 1 cancers-15-04190-f001:**
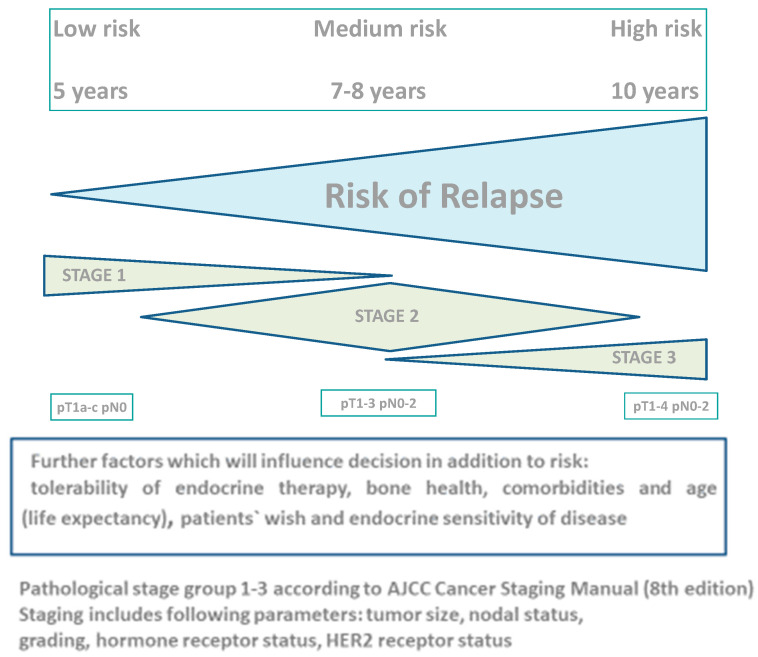
Decision algorithm of duration of adjuvant endocrine therapy.

**Table 1 cancers-15-04190-t001:** Study comparison of extended adjuvant endocrine therapy in early breast cancer.

	Studies	Treated Patients ^1^	Follow-Up (months)	Disease-Free Survival	Overall Survival
5 versus 10 years (tamoxifen)	ATTOM/ATLAS (pre-and postmenopausal) MA-17 ^1^	13,799 5187	108/120 80	sig sig	sig sig (only N+)
5 versus 10 years (aromatase inhibitor)	NSABP B-42 ^1^ MA-17R ^1^	3923 1918	82.8 75.6	not sig sig (DRFS) ^2^ sig	not sig not sig
5–6 versus 7–8 years	DATA ^1^ GIM-4 ^1^	1860 2056	49.2 140.4	not sig (+) ^3^ sig	not sig sig
7–8 versus 10 years	IDEAL ^1^ ABCSG 16 ^1^	1824 3484	79.2 118	not sig not sig	not sig not sig

^1^ ATTOM and ATLAS included pre- and postmenopausal patients; patients included in the other trials were all postmenopausal. ^2^ Distant relapse-free survival. ^3^ Longer treatment better in post hoc analysis in: ER und PgR +, pN+, >/= 2 cm. sig: significant difference in favor of longer therapy.

**Table 2 cancers-15-04190-t002:** Side effects of extended endocrine therapy in different studies.

	Studies	Side Effects (Toxicities)
5 versus 10 years (tamoxifen) 5 years AI vs. placebo after 5 years tamoxifen	ATTOM/ATLAS MA-17	More endometrial cancer, thromboembolic events with longer therapy (ischemic cardiac events reduced) More hormonally related side effects with AI
5 versus 10 years (aromatase inhibitor)	NSABP B-42 MA-17R	More arterial thromboembolic events with longer therapy More bone-related toxic events with longer therapy
7–8 versus 10 years	IDEAL ABCSG 16	No difference in side effects More bone fractures with longer therapy

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
