# Peer review of "Extended Adjuvant Endocrine Therapy in Early Breast Cancer Patients—Review and Perspectives"

_cancers, 2023, doi:10.3390/cancers15164190_

Round 1

Reviewer 1 Report

This manuscript presents an insightful review which discusses extended adjuvant endocrine therapy and the optimum duration of endocrine therapy in early breast cancer. The manuscript is informative and important. However, please consider the following points that might strengthen the manuscript.   Comments: 1. Please provide a title for Table 1. Moreover, what do “+” and “−” mean? Please provide more appropriate words or expressions. 2. Please include a Table showing a summary of the side effects in each trial. 3. The authors propose adjuvant endocrine therapy for 7 years for patients with an intermediate risk and for up to 10 years for patients with a high risk. Please provide a schema of the treatment strategy for use in daily medical practice.

Reviewer 2 Report

This is an excellent review of adjuvant  hormonal therapy in breast cancer. The review is thorough, relevant, but not original. Comprehensive review like this one, helps advance the field and leads to better treatment of patients.
The references are O.K., albeit only until the beginning of 2021.
The table helps to clarify complicated issues. In summary, it is an important paper; however,  very important subjects have been raised recently [such as the use of CDK4-6 in the adjuvant setting and the issue of pregnancy during the years the patient is supposed to receive adjuvant hormonal therapy], that an update may be considered.  The authors may want to update their excellent review. For example, the recent study showing that an interval pregnancy may be considered in women desiring it, should not be missed/

Reviewer 3 Report

This is a comprehensive review of adjuvant endocrine therapy in Luminal A breast cancer. It provides an excellent consensus on current standards and practices in light of very well-reviewed literature featuring the highest level of evidence available. The manuscript is well-written and easy to read and follow. 

The distinction between pre-and post-menopausal patients was made throughout, but it might be more difficult to distinguish for junior and non-specialised surgeons.  A section on endocrine therapy in the neo-adjuvant setting will also add to the manuscript.

The last section: conclusion and further direction, does not offer an insight into work-in-progress or developments in the field of endocrine treatment. It is only a conclusion.

I enjoyed reading this manuscript. Well done!
